# Tubular Cell Cycle Response upon AKI: Revising Old and New Paradigms to Identify Novel Targets for CKD Prevention

**DOI:** 10.3390/ijms222011093

**Published:** 2021-10-14

**Authors:** Letizia De Chiara, Carolina Conte, Giulia Antonelli, Elena Lazzeri

**Affiliations:** Department of Experimental and Clinical Biomedical Sciences Mario Serio, University of Florence, 50139 Florence, Italy; carolina.conte@unifi.it (C.C.); giulia.antonelli@unifi.it (G.A.)

**Keywords:** acute kidney injury, chronic kidney disease, mitotic cell cycle, cell cycle arrest, alternative cell cycle, polyploidy, senescence, fibrosis

## Abstract

Acute kidney injury (AKI) is characterized by a rapid deterioration of kidney function, representing a global healthcare concern. In addition, AKI survivors frequently develop chronic kidney disease (CKD), contributing to a substantial proportion of disease burden globally. Yet, over the past 30 years, the burden of CKD has not declined to the same extent as many other important non-communicable diseases, implying a substantial deficit in the understanding of the disease progression. The assumption that the kidney response to AKI is based on a high proliferative potential of proximal tubular cells (PTC) caused a critical confounding factor, which has led to a limited development of strategies to prevent AKI and halt progression toward CKD. In this review, we discuss the latest findings on multiple mechanisms of response related to cell cycle behavior of PTC upon AKI, with a specific focus on their biological relevance. Collectively, we aim to (1) provide a new perspective on interpreting cell cycle progression of PTC in response to damage and (2) discuss how this knowledge can be used to choose the right therapeutic window of treatment for preserving kidney function while avoiding CKD progression.

## 1. Introduction

Acute kidney injury (AKI), a syndrome characterized by an acute deterioration of kidney function, represents a global healthcare issue [1,2,3]. Many epidemiologic studies demonstrated that AKI survivors frequently develop chronic kidney disease (CKD), which is associated with a high cardiovascular risk and the possibility of progression toward end-stage kidney disease (ESKD) [4]. CKD is a highly prevalent condition that contributes to a substantial proportion of disease burden globally [5]. Yet, there are currently no effective strategies to treat AKI or prevent disease progression to CKD. Given the importance as sites of injury and the fact that they are the predominant cell type in the kidney, proximal tubular cells (PTC) have been the main focus of research on AKI to CKD transition, together with resident fibroblasts, the main source of extracellular matrix production [6,7,8]. PTC are responsible for the reabsorption of 60–70% of water and NaCl, 80% of NaHCO3, and almost all the nutrients present in the glomerular ultrafiltrate. In addition, they maintain the fluid–electrolyte and acid–base balance, ammoniagenesis, Vitamin D synthesis, and immunomodulating functions [9]. PTC can be further divided into those beloging to the S1, S2, and S3 segments, according to their localization along the proximal tubule [10]. PTC display many differences in their cellular ultrastructure and function, and this diversity is translated into a different ability to respond to AKI [10]. Indeed, PTC from the S3 segment of the outer stripe of the outer medulla are the one most affected by ischemic injury [11,12,13]. This region accounts for a unique microvascular environment, which is extremely vulnerable to hypoperfusion, renal hypoxia, and mitochondrial damage [13,14]. Injured PTC trigger chronic inflammation with persistent cytokine production and lymphocyte infiltration [15,16]. In addition, PTC mitochondrial dysfunction has been implicated in tubulointerstitial fibrosis development, increased susceptibility to injury, and CKD progression [17,18,19,20]. Mitochondrial dysfunction also affects the production of 1α,25-dihydroxyvitamin D3, the active form of Vitamin D, recently suggested as a potential beneficial mechanism to modulate and ameliorate the injury response at the tubular level [21]. Studies from the last two decades have tried to elucidate the role of PTC in the intrinsic repair process after AKI [6,7]. In particular, the functional recovery from AKI was traditionally attributed to the regenerative capacity of PTC, which are believed to enter the cell cycle and repair the damage. However, although renal function returns to baseline after an acute insult, AKI survivors frequently develop CKD [22], which is incompatible with the idea that the cell cycle re-entry of PTC repopulates and regenerates the nephron.

To explain this apparent inconsistency, several groups reported that after a severe injury, PTC remain arrested in the G2/M phase of the cell cycle, leading to the so-called maladaptive repair [6,7]. The mechanisms causing this arrest and its precise implications are unknown thus far. Recently, evidence for the existence of alternative cell cycles generating polyploid PTC has further complicated this scenario [23]. This would suggest that the current paradigm about kidney repair mechanisms is oversimplified and requires revision. Dissecting the several cell cycle faces and their dynamics in PTC is crucial to identifying the potentially druggable cell cycle-related molecules or pathways that may be employed for preventing AKI to CKD transition. Collectively, we aim to summarize the major findings on PTC cell cycle behavior in response to AKI, shedding some light on the processes that promote CKD progression and helping to advance our understanding of AKI pathophysiology.

## 2. The Many Faces of Cell Cycle in Tubular Cells after AKI

Distinct cell cycle programming, in response to similar stimuli, are triggered in PTC upon injury, specifically the: (1) mitotic cell cycle, (2) cell cycle arrest, and (3) alternative cell cycle, all of which are strictly controlled by several distinctive molecules that modulate progression through checkpoints (Figure 1).

### 2.1. Mitotic Cell Cycle

The mitotic cell cycle consists of a series of events that, if completed, leads to DNA duplication and the generation of two daughter cells, i.e., cell division or proliferation. The mitotic cell cycle involves four highly regulated phases: G0–G1 phase (cell growth in size), S phase (DNA duplication), G2 phase (check for replication error), and M phase (mitosis). In physiological conditions, the kidney cell turnover is very low and PTC are maintained in G0–G1 phase [24]. Upon ischemic or nephrotoxic damage, all survived PTC are believed to enter the cell cycle, with the attempt to divide and generate new PTC in order to replace lost cells [25,26,27,28,29]. This long-standing model is based on a widespread positivity of PTC for cell cycle markers, such as Ki67, proliferating cell nuclear antigen (PCNA) or bromodeoxyuridine (BrdU) incorporation [26,27,28,29], which would collectively indicate a presumptive great proliferative potential. However, the area most susceptible to the injury is the S3 segment of proximal tubules, where a proliferative response is indeed expected, despite being detected all over the cortex [13]. The observation that even the PTC far from the site of injury successfully enter the S phase of the cell cycle following AKI (i.e., positivity to cell cycle markers) led to the idea that the kidney response to AKI involved all PTC equally [26,27,28,30]. However, the concept that all PTC are endowed with the same ability to proliferate and repopulate the kidney parenchyma is in clear contrast with the fact that AKI survivors frequently develop CKD, a process regarded as maladaptive repair. Accordingly, we (and others) demonstrated that the regeneration of injured tubules involved a specialized intratubular scattered population of renal progenitor cells (RPC) and it is not a prerogative of differentiated PTC [23,31].

RPC were firstly identified and characterized in the human kidney [31,32] and then, through the use of transgenic models [23,33,34], in mice [33,35]. Data obtained from lineage tracing studies (mainly based on the Confetti reporter, see Section 3) have offered unprecedented levels of information regarding organization of resident RPC and their clonal dynamics, providing solid indications on their role in kidney development, homeostasis, and regeneration [23,33,36,37,38,39]. RPC localize mostly to the S3 segment, the one mostly affected by ischemic and nephrotoxic injuries [23,33,35,40,41]. Unlike differentiated PTC, RPC showed an increased resistance to death, following an AKI episode [23]. Remarkably, the use of sophisticated transgenic mouse lines demonstrated that RPC are the sole cells endowed with the ability to successfully complete a mitotic cell cycle and replace lost tubular cells in the necrotic S3 segments of the proximal tubule in injured nephrons [23]. This explains the high proliferation rate observed in this area, which contributes to the structure recovery [23]. Overall, the discovery that kidney harbors a progenitor cell compartment suggests that the massive proliferative response of PTC after AKI is not a regenerative response [42] and raises the question of why differentiated PTC that reside far from the injury site showed widespread positivity for proliferation markers in response to injury.

#### Targeting Mitotic Cell Cycle as Potential Innovative Strategy for CKD Prevention

HDAC inhibitors

Histone deacetylase inhibitors (HDACi) are a relatively new class of compounds involved in epigenetic regulation. Most of what we know about HDACi comes from cancer-focused studies; however, over the past decade, the therapeutic potential of this class of epigenetic regulators in modulating AKI has also been extensively studied, due to their effect in boosting cell cycle progression. Cianciolo Cosentino C. et al. developed a small molecule screen, using zebrafish embryos, to identify compounds that can expand the embryonic RPC pool to enhance recovery and reduce post-injury fibrosis after AKI. Using this approach, they identified a new HDACi, methyl-4-(phenylthio) butanoate (m4PTB), that accelerated recovery when it is administered 24 (mice) to 48 (zebrafish) hours after the initial injury [43]. Accordingly, other two HDACi, trichostatin (TSA) and 4-phenylbutyrate (4-PBA) promoted tubular RPC proliferation by accelerating the intrinsic capacity of the kidney for tubular regeneration upon damage [23]. Their effect resulted from a selective expansion of tubular RPC, restoring tubular cell numbers, reconstituting tubular integrity, and avoiding the development of tissue fibrosis and CKD [23]. These studies provided the first evidence that pharmacological stimulation of RPC proliferation can be proposed as a new strategy to enhance tubular regeneration, leading to a recovery of renal function and avoiding the development of tissue fibrosis and CKD [23,43]. Therefore, drugs that enhance post-injury regenerative responses are particularly attractive because they may still be effective when administered days after the initial insult has occurred (Table 1).

### 2.2. Cell Cycle Arrest

To guarantee the cell integrity during the division process, the mitotic cell cycle is controlled by four checkpoints, i.e., the G1/S, intra-S checkpoint, G2/M, and intra-M checkpoints [44]. Failing to pass one of these checkpoints results in a prolonged arrest in a specific phase or in cell death [45]. Cell cycle arrest may play an important role in the protection of PTC following AKI, by avoiding cell division when they are potentially damaged [46]. Indeed, AKI is often accompanied by the presence of DNA damage [47], which leads to the activation of the ataxia telangiectasia mutated (ATM) and/or ataxia telangiectasia and Rad3-related (ATR) proteins, two phosphaditylinositol 3-kinase family members that phosphorylate several downstream targets, including p53 and checkpoint kinase 2 (CHK2), with consequent production of p21^Waf1/Cip1^, a cell cycle inhibitor that arrests tubular epithelial cells in the G1 phase or in G2/M phase of cell cycle [47,48]. Urinary tissue inhibitor of metalloproteinase-2 (TIMP2) and insulin-like growth factor-binding protein 7 (IGFBP7), both early markers of kidney damage, are closely associated with the G1 cell cycle arrest that occurs during the very early phases of AKI [49,50]. However, although cell cycle arrest has potentially beneficial effects either for repairing DNA damage or avoiding cell division when DNA damage cannot be repaired, if PTC do not re-enter the cell cycle, their prolonged arrest favors the acquisition of a pro-fibrotic phenotype [47,51]. After AKI, a prolonged block in G1 phase caused an increment in expression of transforming growth factor beta (TGFβ), resulting in a senescent cell phenotype, which can potentially lead to the development of fibrosis [52]. Likewise, the arrest in the G2/M phase of cell cycle is associated with the development of fibrosis after an AKI [47]. Yang L. et al. analysed five different mouse models of AKI, and observed that PTC arrested in the G2/M phase produced an increased amount of pro-fibrotic growth factors, such as TGFβ and connective tissue growth factor (CTGF), compared to non-arrested cells [47]. G2/M arrested cells induced the activation of the c-jun NH2- terminal kinase (JNK), an important mediator of the MAPK (mitogen-activated protein kinase) signaling pathway [53]. This persistent activation led to an abnormal production of pro-fibrotic cytokines that was also accompanied by cellular senescence [47,54,55]. Senescent cells influence neighboring cells by producing pro-fibrotic and pro-inflammatory factors, *via* a specific secretome named senescence-associated secretory phenotype (SASP) [56], triggering a vicious circle that eventually led to CKD development. Nevertheless, a recent paper from the McMahon lab, employing single nucleus cell RNA-seq (snRNA-seq) analysis, found a subpopulation of PTC exhibiting pro-inflammatory and pro-fibrotic signature [57], but they did not find any sign of G2/M cell cycle arrest. Interestingly, neither snRNA-seq analysis nor immunofluorescence for phospho-histone H3+ (p-H3+) expression (that is generally used to identify cell cycle arrested cells) identified a G2/M arrest in their pro-inflammatory and pro-fibrotic population of PTC, questioning the actual presence of this response mechanism [57].

#### Targeting Cell Cycle Arrest as Potential Innovative Strategy for CKD Prevention

p21^Waf1/Cip1^

p21^Waf1/Cip1^ protein has a wide spectrum of activities, depending on the cell type and the circumstances of its induction. p21 knockout mice show a more pronounced renal function impairment, and an overall increase in mortality upon AKI [58]. Conversely, p21^Waf1/Cip1^ expression induces G1 arrest, ameliorates AKI and protects PTC against apoptosis in the early phase after AKI, as well as against a further renal insult [59], likely due to the fact that arresting cells in the G1 phase can provide more time for DNA damage repair, avoiding uncontrolled progression toward cell death. *Vice versa*, early tubular cell mitosis can potentially lead to mitotic catastrophe, which deletes cells with DNA damage. The protective effect of p21^Waf1/Cip1^ in AKI is also associated with its ability to bind and inhibit CDK2. In vivo pharmacological inhibition of CDK2 resulted in less severe nephrotoxicity after cisplatin treatment [60]. Additionally, the administration of other CDK inhibitors, such as CDK4/6, before AKI significantly improved kidney function 24 h after injury, despite a reduced proliferation of PTC, which were instead arrested in G1 phase [42]. However, in addition to the beneficial effects on PTC, p21^Waf1/Cip1^ may also play a role in driving the progression to CKD by inducing TGFβ production, ultimately leading to fibrosis [61,62]. Accordingly, in a renal ablation model, a lack of p21^Waf1/Cip1^ diminished cell cycle arrest, avoiding long-term renal dysfunction and interstitial fibrosis [61]. However, a recent novel p21^Waf1/Cip1^ deficient mouse line showed exacerbation of fibrosis after damage [63]. The finding that p21 has bimodal roles, protecting (in the acute models) or accelerating the progression of fibrosis (in models of chronic renal failure) suggest a differential role p21^Waf1/Cip1^ related to the stage of damage (acute vs. chronic). Moreover, upstream pathways for p21 activation, such as Smad7, are being investigated for their protective role in AKI. In a Smad7 knockout mouse model, more severe renal impairment, including higher levels of serum creatinine and massive tubular necrosis, was developed at 48 h after AKI. Mechanistic studies revealed that more severe AKI in Smad7 knockout mice was associated with a marked activation of TGFβ/Smad3- p21^Waf1/Cip1^ signaling, thereby inducing a sustained G1 cell cycle arrest that is responsible for the development of fibrosis [64]. Collectively, p21^Waf1/Cip1^ can have both beneficial and detrimental effects according to the timing and the duration of the cell cycle arrest driven by it (Table 1).

p53

Genetic and pharmacological inhibition of p53 has been shown to modulate kidney repair after AKI, attenuating the massive apoptotic and necrotic death of tubular epithelial cells and acute kidney failure [65,66]. Conversely, p53 activation is required for the transcription of pro-fibrotic genes in PTC in a mouse model of unilateral ureteral obstruction (UUO) and, thus, involved in CKD progression [67]. Although the protective effects of p53 inhibition in the acute phase of AKI have been extensively reported, very little is known about the impact of acute p53 inhibition on the chronic *sequelae* after AKI. Inhibition of p53 with pifithrin-α at the time of injury up to a week showed no protective effect on development fibrosis at later time points [68]. However, Yang L. et al. showed that a late inhibition of p53 with pifithrin-α (on day 3 and 14) after ischemic reperfusion injury prevented the development of renal fibrosis [47]. In addition, p53 is negatively regulated by the interaction with the MDM2 protein [69], implying the MDM2-p53 pathway as another possible therapeutic target to prevent CKD progression upon AKI. MDM2 inhibition shows a dual effect in the AKI response. Treatment with MDM2 inhibitors impairs re-epithelialization as part of the repair process of tubular cells after AKI, but also prevents tubular necrosis by promoting cell cycle arrest and DNA repair in the early phase after AKI [70,71,72]. These evidence suggest that pharmacologic inhibition of p53, if appropriately managed, may have significant clinical implications. Nevertheless, it is important to emphasize that ~50% of human cancers harbor p53 deletions and mutations and p53 deficiency in mice is associated with a high frequency of spontaneous cancers [73]; therefore, treatments that interfere with p53 activity should be used only with great caution and for a limited time interval [45,74] (Table 1).

HDAC inhibitors

HDAC inhibitors have been extensively investigated, due to their effect in reducing G2/M arrest of surviving PTC after AKI, when administered late after the initiation of renal injury [75]. When m4PTB treatment is delayed 4 days after the initiating injury accelerates recovery, improving renal function and reducing fibrosis in a model of aristolochic acid induced-AKI in mice [76]. Moreover, the phenylthiobutanoic acids (PTBA) prodrug UPHD186 also accelerates recovery and reduces post-injury fibrosis after ischemia when administered several days after AKI. However, whereas delayed treatment (96 h after injury) improved survival and renal histology and decreased development of fibrosis, an early treatment (48 h after injury) further worsened kidney damage [77]. Blockade of HDAC6 with ACY-1215 successfully alleviates the development of renal fibrosis in a UUO model of fibrosis [78]. Thus, developing successful therapeutic treatments to manage AKI is complicated by biological variables, such as the cell types targeted by these compounds and time-dependency, as both may affect therapy efficacy. However, the observation that post-injury treatment enhances AKI recovery in different experimental systems represents a significant preclinical advance in treating human AKI. In particular, a delayed HDACi treatment provide a theoretical basis for future clinical trials to prevent and treat renal fibrosis in patients presenting with AKI [76] (Table 1).

### 2.3. Alternative Cell Cycles

Endoreplication is an evolutionarily conserved alternative cell cycle program, during which cells replicate their genomes without cytokinesis, resulting in polyploid cells [79,80]. Endoreplication can be further divided into endocycle and endomitosis. During endocycle, the cells oscillate between G1 and S phase completely skipping mitosis and resulting in mononuclear polyploid cells. In endomitosis, the cells enter the G2/M phase without nuclear division, leading to the formation of mononuclear polyploid cells or with nuclear division, leading to the formation of multinucleated polyploid cells [40]. In spite of the great proliferative potential attributed to the differentiated PTC, we and others have recently described that PTC respond to AKI by triggering polyploidization-mediated hypertrophy but do not actively proliferate [23,81]. Polyploid PTC are mainly localized in the cortex and detected 30 days after AKI, while healthy kidneys display a relatively low percentage of polyploid PTC, suggesting that PTC polyploidy may be a stress-related mechanism in response to AKI. Crucially, in the kidney polyploid PTC are frequently mononuclear making it impossible to distinguish them from their diploid counterparts unless by combining techniques that permit measurement of DNA content and detection of cell cycle phases simultaneously [23,81] (see Section 3). This has likely delayed the recognition of polyploidy in the kidney. Moreover, the sole analysis of the DNA content cannot distinguish the polyploid PTC from the G2/M arrested ones (see Section 3), implying that cell cycle arrested PTC may be instead polyploid PTC (see Section 3). Importantly, in cultured human primary proximal tubular epithelial cells (RPTEC) infected with HIV [82], a prolonged G2 triggered by virus infection led to either mitotic cell death, due to extra centrosomes or to polyploidization, suggesting that polyploidy may be a means to escape death. Likewise, tubular cells exhibiting mutations in genes involved in DNA damage repair are more susceptible to DNA damage and genome instability displaying an increased DNA content [83,84]. This leads to the development of tubular atrophy and fibrosis [83,85]. As propagating a stable genome is part of a regular mitotic cell cycle progression, the corollary may be that alternative cell cycle program and genome instability are linked conditions. These observations implicate polyploidization as both an adaptation to genotoxic stress, and/or a trigger of it.

#### Targeting Alternative Cell Cycle as Potential Innovative Strategy for CKD Prevention

YAP1

The evolutionary conserved Hippo pathway plays a pivotal role in controlling cell growth during development and regeneration and its dysregulation is extensively implicated in various cancers. Central to the Hippo signaling cascade is the transcription factor YAP1 (yes-associated protein 1). Given the prominent role of YAP1 in cell and tissue growth, it was recently suggested that YAP1 is able to modify cell cycle progression to accommodate tissue growth [86]. Indeed, Kim W. et al identified APC/C^Cdh1^, a core component of cell cycle control machinery, as an evolutionarily conserved regulator of large tumor suppressor (LATS) kinases, which inhibit YAP1 activation. Particularly, APC/C^Cdh1^ destabilizes LATS1/2 kinases in the G1 phase of cell cycle, leading to increased YAP1 activity promoting G1/S transition by upregulating downstream gene expression, including E2F1, a critical controller of endoreplication [87]. *Drosophila* CDH1-homolog fizzy-related (FZR1) protein dictates the decision between mitosis and endoreplication [88], suggesting a possible link between endoreplicating process and YAP1. Interestingly, YAP1 expression has been shown to associate with chronic inflammation, fibrosis, and functional loss but its role and mechanism in AKI to CKD transition remains unclear [89,90,91,92,93,94]. Following AKI, YAP1 is persistently activated and associated with fibrosis and CKD development. In addition, YAP1 activation is found to be triggered by TGFβ1, one of the most important cytokines regulating fibrosis deposition [95]. Despite that, selective proximal tubule YAP1 deletion exacerbated kidney damage after AKI and delayed functional recovery and kidney repair [96]. Conversely, inhibiting YAP1, after the acute phase of AKI, attenuated renal function decline and interstitial fibrosis, further suggesting that the sustained activation of YAP1 in the post-acute phase of AKI is involved in CKD progression [93,94]. Collectively, these observations implicated YAP1 as a possible therapeutic target to favor kidney repair, if modulated within the correct window of opportunity (Table 1).

**Table 1 ijms-22-11093-t001:** Therapeutic strategies based on cell cycle targeting for CKD prevention. IRI: ischemia reperfusion injury; KO: knock-out.

	Cell Cycle Phase	Target	Therapeutic Strategy	Timing of Treatment	Effect	Reference
	G1/S		HDAC inhibitor (4-PBA)	1 day after IRI	Renal function recovery and tubular regeneration	[23]
**Mitotic cell cycle**	G1/S	HDAC	HDAC inhibitor (TSA)	1 day after IRI	Renal function recovery and tubular regeneration	[23]
	G1/S		HDAC inhibitor (m4PTB)	1 day after injury	Renal function improvement and fibrosis decrease	[43]
	G1	p21	p21(−/−) mouse, constitutive KO	-	Increase of tubular cell death and mortality	[58]
	G1		p21(−/−) mouse, constitutive KO	-	No fibrosis development	[62]
	G2		p21(−/−) mouse, constitutive KO	-	Fibrosis exacerbation	[63]
	G1		Cdk2 inhibitor (Purvalanol)	1 day after injury	Nephrotoxicity reduction	[60]
	G1		Cdk4/6 inhibitor (PD 0332991)	1 h before IRI	Renal inflammation attenuation and kidney damage improvement	[42]
**Cell cycle arrest**	G1		Smad7(−/−) mouse, constitutive KO	-	Tubular regeneration impairment	[64]
	G2/M		p53 inhibitor (Pifithrin-a)	On the day of IRI	Fibrosis increase	[68]
	G2/M	p53	MDM2 antagonist (Nutlin-3a)	1 day before injury	Renal inflammation and tubular injury decrease	[71]
	G2/M		p53 inhibitor (Pifithrin-a)	3 days after injury	Fibrosis decrease	[47]
	G2/M		JNK inhibitor (SP600125)	7 days after injury	Fibrosis decrease	[47]
	G1/S	HDAC	HDAC inhibitor (UPHD186)	3 days after injury	Fibrosis decrease	[75]
	G1/S		HDAC inhibitor (m4PTB)	4 days after injury	Fibrosis decrease	[76]
	G1/S		HDAC inhibitor (UPHD186)	4 days after injury	Fibrosis decrease	[77]
	G1/S		HDAC6 inhibitor (ACY-1215)	On the day of injury	Fibrosis decrease	[78]
**Alternative cell cycle**	G1/S	YAP1	YAP1(−/−) mouse, renal conditional KO	-	Delay of renal function recovery	[96]
	G1/S		YAP1 inhibitor (Verteporfin)	On the day of IRI	Delay of renal function recovery	[96]
	-		YAP1 silencing (Ad-shYAP)	7 days after IRI	Renal function recovery and fibrosis decrease	[93]
	-		KLF4 silencing (Ad-shKLF4)	7 days after IRI	Renal function recovery and fibrosis decrease	[93]
	-		KLF4 overexpression (Ad-KLF4)	7 days after IRI	Fibrosis increase	[93]
	G2/M		YAP1 inhibitor (Verteporfin)	3 days after IRI	Renal inflammation and fibrosis decrease	[94]

The Notch pathway

Among the pathways that control the switch from mitotic cell cycle to alternative cell cycle, the Notch pathway is one of the most important. As different organs have different layers of regulation, it is likely to assume that various pathways will result in tissue-specific cell cycle responses. Similar to the Hippo pathway, the Notch pathway is a highly conserved signaling pathway. Notch itself is a cell-surface receptor that transduces signals upon ligand binding, which lead to cleavage and release of the Notch intracellular domain (NICD), in order to regulate transcriptional complexes [97]. In the Drosophila nervous system, the Notch signaling pathway is crucial for the endocycle versus endomitosis choice [98], while it controls the switch from mitotic to alternative cell cycle in Drosophila follicle cells [99]. In a Drosophila Notch-driven model of a solid-tumor, Wang XF. et al. found that the tumor-initiating cells undergo endoreplication to become polyploidy [100]. The upregulation of Notch signaling induces these polyploid cells to re-enter mitosis and undergo tumorigenesis. Interestingly, mice that overexpressed NICD1 in all tubular epithelial cells presented a progressive decline of kidney functionality, indicative of CKD, suggesting the presence of alternative cell cycles [101]. Indeed, Notch1 expression in renal tubular cells was found to be both necessary and sufficient for the development of tubulointerstitial fibrosis, while its deletion reduced fibrosis development [102]. Notch1 overexpression also correlated with increased expression of Cyclin-A1, Cyclin-D1, and Cyclin-E1 [102], the latter being a key controller of endoreplication [103]. In vivo studies employing pharmacologic inhibition to reduce Notch signaling activation in the setting of AKI showed mixed outcomes. Huang R. et al. found that the treatment ameliorated the severity of tubular damage after AKI in rats [104], whereas Chen J. et al. showed that the same treatment delayed functional recovery after AKI in mice [105]. However, as none of the groups disclaimed the time point at which the inhibitor was administered, the outcome may have been influenced by a different choice of timing for the treatment. All together, the blockade of Notch signaling may represent a novel therapeutic strategy to prevent CKD development (Table 1).

## 3. Mitotic Cell Cycle, Cell Cycle Arrest, or Alternative Cell Cycle: How Can We Get to the Bottom?

Given the many faces of kidney injury response, a major challenge is to faithfully distinguish between different cell cycle events. However, all the main methodologies employed so far have specific shortcomings that undermine definitive conclusions. Collectively, we summarize old and new strategies employed to conclusively detect a complete mitotic cell cycle, resulting in cytokinesis and generation of two diploid daughter cells versus cell cycle arrested cells or alternative cell cycle generating polyploid cells (Figure 2).

### 3.1. Cell Cycle Markers

Cell cycle labelling is based on the expression of traditional markers which allow to discriminate the different phases accordingly. In the past, these markers have been widely used to demonstrate cell cycle entry of PTC, in response to kidney injury [26]. However, cell cycle entry does not necessarily culminate in cell division. Consistently, recent data showed that PTC enter the cell cycle increasing their DNA content but do not undergo cell division (i.e., alternative cell cycle or cell cycle arrest) staining positive for cell cycle markers, although they do not originate two daughter cells [23,47,81]. S phase markers like PCNA, 3H-thymidine, 5-ethynyl-20-deoxyuridine (EdU), or BrdU, do not distinguish mitotic cell division from cell cycle arrest or alternative cell cycle, as the cells will enter the S phase, expressing the cell cycle markers in all three conditions. Likewise, the G2/M phase marker, p-H3, does not differentiate among mitotic cell division, cell cycle arrest, or alternative cell cycle, as the chromosomes condense during metaphase and are readily labeled with p-H3 in all cell cycle events [106]. The Ki67 marker labels all phases of the cell cycle and only distinguishes cycling cells from non-cycling cells [107]. Overall, cell cycle markers are not reliable indicators of cell division, causing an overestimation of proliferating response after injury. However, limitations on the use of these markers have not been considered until recent innovative strategies have been developed.

### 3.2. DNA Content Analysis

Traditionally, the discrimination of cell cycle events has been largely investigated through the measurement of DNA content (ploidy), in order to distinguish between diploid (2C) and polyploid (>4C) cells. DNA content analysis can be performed with several methods: (1) by flow cytometry [108]; (2) by single-cell DNA sequencing [109]; and (3) by imaging fixed cells [110]. DNA content analysis has been conventionally performed through flow cytometry and has, more recently, taken advantage of whole-genome sequencing, by using a method to detect copy number alterations in single cells and inferring the copy number through sequencing read depth [109]. An additional method to quantify DNA content has also been reported by Losick V. et al., by staining the Drosophila ovary and follicle cells with 40,6-diamidino-2-phenylindole (DAPI) to measure the signal within each nuclear boundary and quantifying the cell ploidy [98]. From a technical point of view, the main issue is represented by the fact that G2/M cycling cells or G2/M arrested cells and polyploid cells, having the same amount of DNA (4C), are indistinguishable based solely on the DNA content [26,29,111].

### 3.3. Lineage Tracing with Fluorescent Reporters

DNA content measurement combined with the simultaneous analysis of cell cycle live imaging of fluorescent reporters allows us to accurately detect cell cycle events in vivo. This strategy is based on fluorescent ubiquitination-based cell cycle indicator (FUCCI2aR) technology, which distinguish nuclei of cells in the G1 phase, expressing the fluorescent protein mCherry fused with a truncated human Cdt1 (hCdt1), from nuclei of cells in S/G2/M phase, expressing the fluorescent protein mVenus fused with the 110 amino acid N-terminus of the human Geminin (hGeminin) [112,113]. The first one accumulates during G1 phase and is degraded at the G1/S transition. The second one accumulates during S/G2/M phases and is rapidly degraded prior to cytokinesis. Cells also appear as yellow at the G1/S boundary [112,113]. As a result of this technology, cell cycle phase analysis performed, combined with the measurement of DNA content, can distinguish between proliferating/arrested cells (diploids in G2/M with 4C DNA content), which express mVenus protein, and polyploid cells having a DNA content ≥ 4C, which express mCherry protein (stationary in G1) [23,81,112,113]. This technology, however, does not allow to distinguish between proliferating and G2/M arrested cells (Figure 2).

Recently, an elegant genetic system has been established to unequivocally identify polyploid cells and trace their fate in vivo, using the multicolored reporter Confetti [114,115]. The Confetti allele consists of a floxed stop cassette, followed by four different reporter genes: the nuclear green fluorescent protein (GFP), the cytoplasmic yellow fluorescent protein (YFP), the cytoplasmic red fluorescent protein (RFP), and the membrane-bound cyan fluorescent protein (CFP). After Cre recombination, only one fluorescent protein is stochastically expressed from each Confetti allele. The Confetti reporter has been extensively used for in vivo experiments focused on lineage tracing to evidence clonal expansion [23,33,116,117]. The possibility to tag a single desired cell with a reporter and identify all the progeny derived from this cell has made the Confetti system extensively used [115]. Recently Matsumoto T. et al. described an innovative application of the Confetti reporter. In heterozygous-Confetti mice, one of the two sets of chromosomes harbors a Confetti allele, and thus, diploid cells can express only one fluorochrome, resulting in monocolored cells. On the contrary, polyploid cells carrying two or more sets of chromosomes can activate two or more fluorochromes at the same time in the same cell, resulting in multicolored cells. Unlike the FUCCI2aR technology, this revolutionary use of the Confetti reporter allowed the localization of polyploid cells within a tissue, without the need to combine it with DNA content analysis, and helped to shed light on the role of polyploid cells and ploidy dynamics in liver response upon injury [114,115].

## 4. Biological Significance of PTC Cell Cycle Behavior

Cell cycle behavior of PTC plays a crucial role in determining the kidney response to injury and the renal functional outcome after AKI. The studies summarized in this review, have highlighted how a “traditional” cell cycle, which ends with a mitotic event, is likely the prerogative of the progenitor compartment, while the majority of PTC enter an alternative cell cycle, which leads to polyploidization, or undergo cell cycle arrest, driving AKI to CKD transition. The concept that only progenitor cells can undergo mitotic cell cycle, resulting in an actual proliferation, is also supported by recent papers, which have proved that kidney cancers derive from the expansion of a single progenitor cell [101,118]. Indeed, endowing all PTC with the ability to undergo proliferation, hence the potential to accumulate mutations, is counter-productive, as it increases the chances of developing tumors over the course of a lifetime. Evolutionarily, polyploid/arrested PTC may represent a buffering mechanism to prevent tumor development [119]. In line with this concept, the heart, that is mostly composed by polyploid cells, has notoriously a very low incidence of cancer development [120]. However, cell cycle exit is not a universal feature of polyploid cells [79] and proliferation combined with ploidy reduction was shown to be an early step in the initiation of carcinogenesis in polyploid hepatocytes [114,115]. Collectively, understanding how ploidy variations affect PTC function is essential to understand their role in the response to kidney injury. For instance, the existence of polyploid cells in different organs, such as the liver and heart, suggests that polyploidy might have specific physiological advantages that have not yet been fully elucidated [79,121]. Accordingly, as an adaptive mechanism to injury, polyploidy offers several advantages, including rapid adaptation to stress [79], compensation for cell loss [122], enhanced cell function [123] and protection against cancer development [119]. Although the biological significance of this process remains completely unreported in the kidney, polyploidization of PTC could represent a crucial adaptive stress response. In the kidney, when the workload presented to the organ increases, its functional response is proportionately augmented. Yet, the number of nephrons is determined early in life, and no matter how high the demand, that number of functional units does not increase [23,124,125,126,127]. In this context, PTC polyploidization can represent a means to proportionally increase the kidney’s functional response to cope with an increased organ workload. However, both polyploidization and cell cycle arrest, have considerable trade-offs that affect long-term outcomes after an AKI episode [40,128,129]. Indeed, the association of tissue polyploidization, fibrosis, and senescence, has been demonstrated for the liver [130,131], heart [131,132], and is likely to account also for the kidney [90]. Since cells arrested in G2/M phase are indistinguishable from polyploid PTC with a ≥4C DNA content arrested in G1, when measuring only the DNA content [26,29,111], these polyploid PTC could be reminiscent of cells arrested in G2/M phase, previously described by several groups as drivers of CKD progression [47,63,133,134]. Indeed, as cells become arrested in G2/M phase to provide more time for DNA damage repair, avoiding cell cycle progression of damaged cells; the key question would be why these arrested cells do not either progress toward the cell cycle or undergo apoptosis but stay arrested indefinitely. Post-AKI fibrosis and senescence could, therefore, be a consequence of cell cycle arrest upon polyploidization in concert with G2/M cell cycle arrest, a process currently named “maladaptive repair” [24,135].

## 5. Future Perspectives

Injury and death of tubular cells are recognized as the main factors driving the pathogenesis of AKI. Upon kidney damage, the regenerative capacity of RPC is limited, and complete loss of RPC compartment disrupts regeneration of an affected tubule segment, which can result in irreversible loss of the nephron. Moreover, excessive polyploidization or cell cycle arrest are associated with fibrosis and senescence, i.e., CKD. In vivo experiments have highlighted that modulation of cell cycle progression may have both beneficial and/or detrimental effects on AKI recovery and CKD progression [61,93,94]. This apparent inconsistency is likely related to biological variables, such as the cell type that is targeted by that specific compound and the time-dependency of treatments, which may affect treatment efficacy. Indeed, drugs that arrest PTC in G1 phase of cell cycle, protecting them against apoptosis and/or drugs that promote regeneration of RPC by enhancing their ability to proliferate or increasing their survival capacity are the sole compounds that work when administered immediately after damage. Conversely, drugs that act on G2/M arrest or polyploidization show very different effects, depending on the timing of administration. The inhibition of both mechanisms in the early recovery phase after AKI leads to impaired recovery and increased mortality [58,96]. On the contrary, the inhibition after the acute phase of damage, ameliorated fibrosis development and prevented AKI to CKD transition [61,93,94]. Overall, drugs that act on PTC have a favorable effect only when administered after the acute phase of damage, in accordance with the hypothesis that polyploidization and G2/M arrest are required to sustain kidney function, while drugs that act on the progenitor compartment have a beneficial effect also when administered immediately after damage (Figure 3). These divergent results suggest that the role of cell cycle progression in AKI and its modulation is complicated and still incompletely understood. Early support of cell preservation mechanisms (cell cycle arrest or polyploidization) may determine protection of PTC. Conversely, once the DNA damage is repaired and the renal function is recovered, it may be crucial to rapidly reverse these processes so that the adverse consequences (fibrosis and senescence) are avoided [48]. The adaptations that occur in response to the irreversible loss of tubular cells may ensure short-term survival, but they have considerable trade-offs that affect the long-term outcomes after an AKI. In conclusion, a better understanding of cell cycle events and their manipulation could be of great value in the development of successful therapeutic strategies, helping to choose the right window of opportunity to preserve kidney function and promote regeneration, while avoiding CKD progression (Figure 3).

## 6. Conclusions

In this review, we aimed to describe and interpret recent data relating to the role of cell cycle response during AKI and in the following CKD progression. In particular, we attempted to highlight the relevance of choosing the right timing for treatment administration, in relation to the expected therapeutic outcome. Currently, AKI and CKD are still largely treated as separate syndromes. Nonetheless, it is increasingly recognized that AKI and CKD are closely linked and likely promote one another [4]. However, the effective treatment of AKI patients, as well as risk-stratification of those who will progress toward CKD, has been unsuccessful thus far. Considering the growing number of AKI survivors, it is apparent that a better understanding of the repairing and pathological mechanisms involved in AKI to CKD transition is pivotal, in order to speed up the development of the next-generation therapies to treat AKI patients.

## Figures and Tables

**Figure 1 ijms-22-11093-f001:**
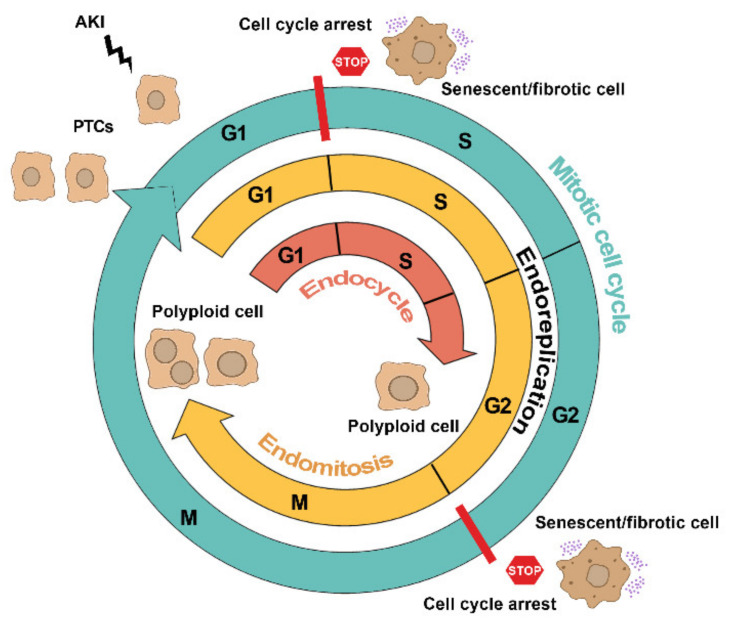
Distinct cell cycle programming triggered in PTC upon acute injury. The scheme depicts the different possible fates of cell cycle programming in PTC after AKI. The mitotic cell cycle (light blue line) progressing through G1, S, G2, and M phases generates two daughter cells. Cell cycle arrest (red line with stop sign) of PTC at G1 and G2 checkpoints triggers a senescent/fibrotic phenotype. Alternative cell cycle (i.e., endoreplication) generates mono/multinuclear polyploid cells via endomitosis (yellow line) or mononuclear polyploid cells via endocycle (red line).

**Figure 2 ijms-22-11093-f002:**
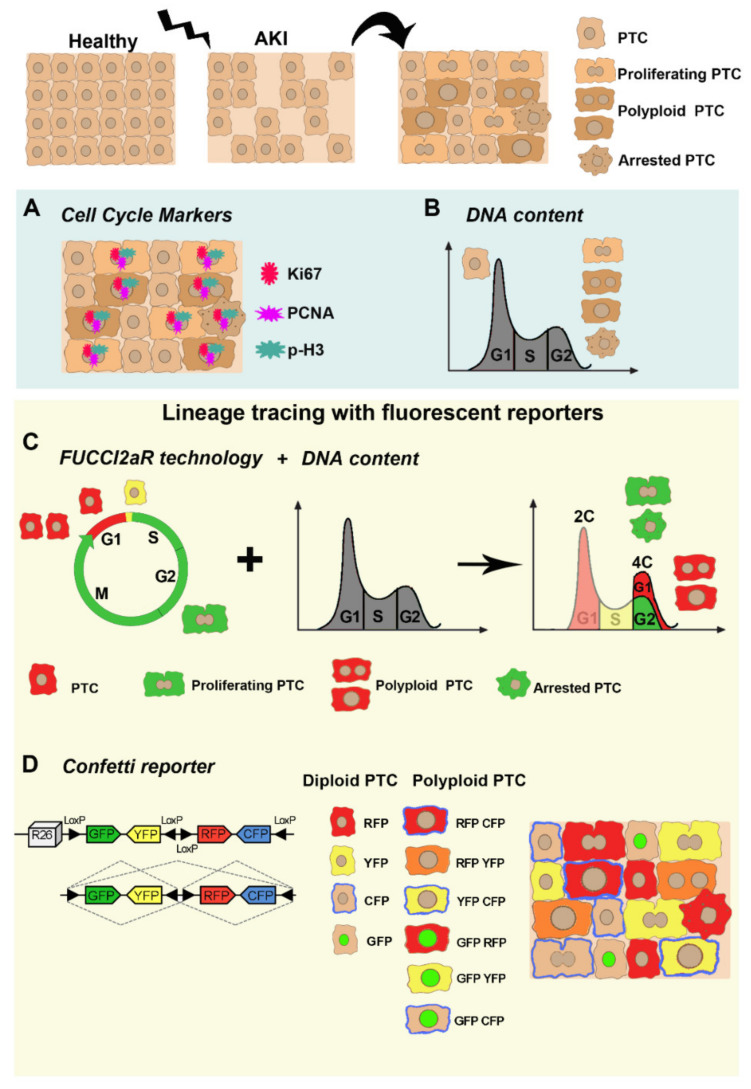
Old and new strategies employed to study cell cycle behavior of PTC after acute injury. PTC respond to acute injury by triggering different strategies: mitotic cell cycle, cell cycle arrest and alternative cell cycle. Use of cell cycle markers (**A**) and DNA content analysis (**B**) (light blue box; old strategies) cannot distinguish among the proliferating PTC (generated by “traditional” mitotic cell cycle) and the arrested PTC or the polyploid PTC (generated by alternative cell cycle). Recently, sophisticated strategies (light yellow box, new strategies) are now being employed to detect proliferating PTC, completing a mitotic cell cycle versus polyploid PTC generated by endoreplication. (**C**) The FUCCI2aR technology coupled with the measurement of the DNA content, can discern between PTC (diploids in G2/M), which express the mVenus protein (green cells), from polyploid PTC having a DNA content ≥ 4C, which express mCherry protein (red cells). However, this technology does not allow to discriminate proliferating PTC from arrested PTC because they are both diploids in G2/M expressing mVenus protein (green cells). (**D**) Innovative application of the Confetti reporter allows the recognition of diploid PTC, which express only one color (YFP, RFP, CFP, and GFP) of the reporter cassette from polyploid PTC, which express two or more combinations of color in the same cell (RFP-CFP, RFP-YFP, YFP-CFP, GFP-RFP, GFP-CFP, and GFP-YFP). This technology, however, does not allow the identification of arrested cells.

**Figure 3 ijms-22-11093-f003:**
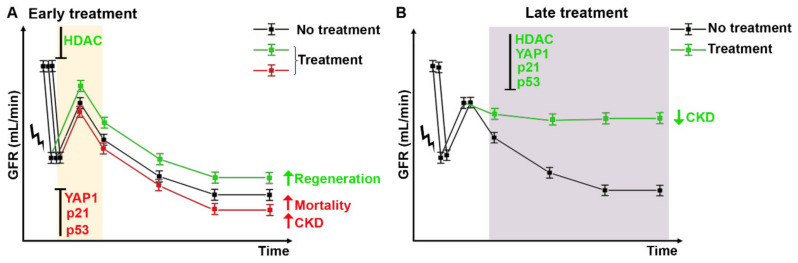
Windows of opportunity for potential druggable targets to treat AKI and avoid CKD progression. (**A**) Early treatment after AKI with HDAC inhibitors enhance kidney regeneration (green line) by promoting RPC proliferation. Conversely, early treatment with drugs that inhibit cell cycle arrest and/or polyploidization (p21, p53, and YAP1 inhibitors) increase mortality of mice and lead to CKD (red line). (**B**) Late treatment, after the acute phase of damage, with drugs that inhibit cell cycle arrest and/or polyploidization (p21, p53, HDAC, and YAP1 inhibitors) ameliorates kidney function and prevents fibrosis and senescence (green line). GFR: glomerular filtration rate.

## Data Availability

Not applicable.

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
