# Peer review of "Tubular Cell Cycle Response upon AKI: Revising Old and New Paradigms to Identify Novel Targets for CKD Prevention"

_ijms, 2021, doi:10.3390/ijms222011093_

Round 1

Reviewer 1 Report

The manuscript by De Chiara and colleagues is a comprehensive review on a very interesting topic. The content is well presented however the following points should be considered:

1) Section 1. Introduction section should be modified substantially and expanded. Include separate section (line 36) describing the tubular cell structure and function in detail. Readers would also benefit from a figure.

2) Move figures closely to each section they refer to for example Figure 1.

3) Authors should consider describing the cell cycles in one section and the therapeutic targets in a separate section.

4) Section 2.1 Expand on scattered RPCs. Include paragraphs.

5) Include a Table summarizing the therapeutic strategies on all cycles.

6) Include a separate conclusion section.

Minor comments:

1) Carefully proofread manuscript and correct for both language editing as well as typos. eg lines 183,328.

2) Lines 457-459. Incomprehensible sentence. Reword.

3) Lines 467-469. Add references on both prevention of AKI to CKD transition and increased mortality.

Author Response

Reviewer 1

The manuscript by De Chiara and colleagues is a comprehensive review on a very interesting topic. The content is well presented however the following points should be considered.

1) Section 1. Introduction section should be modified substantially and expanded. Include separate section (line 36) describing the tubular cell structure and function in detail. Readers would also benefit from a figure.

We thank the reviewer for the useful suggestions. As requested, we have expanded the introduction section with a detailed description of the tubular cell structure and function (line 37-53). Although we agree with the reviewer regarding the beneficial value of a figure, we felt that adding a new figure on tubular cell structure and function is beyond the scope of this review, which is focused on cell cycle behavior of tubular cells, resulting in an overload of information.

2) Move figures closely to each section they refer to for example Figure 1.

Done.

3) Authors should consider describing the cell cycles in one section and the therapeutic targets in a separate section.

We thank the reviewer for the suggestion. We have thoroughly discussed the possibility to move the therapeutic targets in a separate section but ultimately we felt that it would have weaken the take home message that the different compounds act on different cell cycle-related processes and so they should be administered accordingly.

4) Section 2.1 Expand on scattered RPCs. Include paragraphs.

We have expanded the section describing the RPC as suggested by the reviewer (lines 109-117, highlighted in red). However, we did not include a separate paragraph as a throughout description of RPC is beyond the scope of this review.

5) Include a Table summarizing the therapeutic strategies on all cycles.

Done.

6) Include a separate conclusion section.

We added a conclusion section.

Minor comments:

1) Carefully proofread manuscript and correct for both language editing as well as typos. eg lines 183,328.

We fixed the typos and the typing errors throughout the manuscript.

2) Lines 457-459. Incomprehensible sentence. Reword.

We reworded the sentence (lines 485-487).

3). Lines 467-469. Add references on both prevention of AKI to CKD transition and increased mortality.

References have been added (ref. 58, 96, 61, 93 and 94).

Reviewer 2 Report

The authors did an intensive literature review on the selected topic. The text is generally well written and comprehensive to the readers. However, the manuscript is overloaded with too much information and often distracts from the important contents, so the content in the paragraphs in general should be prioritized (e.g the study of HDACi on zebrafish could be a single sentence, instead of a long paragraph).

Line 87, “see paragraph 3”, this misleads, so the author should consider changing to “see section 3”

The author should avoid conversation type of text (line89).

Minor language checks are needed.

Author Response

Reviewer 2

The authors did an intensive literature review on the selected topic. The text is generally well written and comprehensive to the readers. However, the manuscript is overloaded with too much information and often distracts from the important contents, so the content in the paragraphs in general should be prioritized (e.g the study of HDACi on zebrafish could be a single sentence, instead of a long paragraph).

We agree with the reviewer. The text has been modified according to the reviewer suggestion.

Line 87, “see paragraph 3”, this misleads, so the author should consider changing to “see section 3”

We agree with the reviewer. Paragraph has been substituted with section throughout the manuscript.

The author should avoid conversation type of text (line89).

We apologies for the mistake. We have removed all the conversation type of text throughout the manuscript.

Minor language checks are needed.

We fixed the typos and the typing errors throughout the manuscript.

Reviewer 3 Report

The aim of the paper seems to be to review the tubular cell cycle response upon acute kidney injury analysing novel targets for chronic kidney disease prevention. Overall, a comprehensive review of an important issue.

The article is very detailed and descriptive, and I enjoyed reading it and learning from it.

The topic is of great interest in the scientific community, the work is well organized and comprehensively described and the references are up to date. I also really appreciated the future perspective section.

I support the publication of this paper, but several aspects of this paper should be improved:

  • I suggest you improve the introduction section mentioning the role of vitamin D in tubular homeostasis as inflammation mediator, by the Megalin-Cubilin-Amnionless and FGF23-Klotho axis, you can find more information in the article “Gembillo G et al Protective Role of Vitamin D in Renal Tubulopathies. Metabolites. 2020 Mar19;10(3). pii: E115. doi: 10.3390/metabo10030115” and the role of vitamin d as immunomodulator in glomerulonephritis (see Gembillo G, Siligato R, Amatruda M, Conti G, Santoro D. Vitamin D and Glomerulonephritis. Medicina (Kaunas). 2021 Feb 22;57(2):186. doi: 10.3390/medicina57020186. PMID: 33671780; PMCID: PMC7926883).
  • When you write about TGF-β, you should mention the relationship between AKI, elevated TGF-β and upregulation of the yes-associated protein (YAP)/transcriptional coactivator with PDZ-binding (TAZ) pathway, you can find more in the paper “Anorga S, Overstreet JM, Falke LL, Tang J, Goldschmeding RG, Higgins PJ, et al. Deregulation of Hippo-TAZ pathway during renal injury confers a fibrotic maladaptive phenotype. FASEB J. 2018;32(5):2644–57” and “Liang M, Yu M, Xia R, Song K, Wang J, Luo J, et al. Yap/Taz Deletion in Gli(+) Cell-Derived Myofibroblasts Attenuates Fibrosis. J Am Soc Nephrol. 2017;28(11):3278–90”.
  • Another aspect that should be improved is the mitochondria dysfunction in AKI damage, you can find more in the paper “Yu SM, Bonventre JV. Acute kidney injury and maladaptive tubular repair leading to renal fibrosis. Curr Opin Nephrol Hypertens. 2020 May;29(3):310-318. doi: 10.1097/MNH.0000000000000605. PMID: 32205583; PMCID: PMC7363449” in the paragraph “Mitochondria Dysfunction”.
  • A conclusion section should be added.
  • An abbreviations section should be added.
  • A minor English check should be provided.
  • The resolution of the images should be improved.

Author Response

Reviewer 3

The aim of the paper seems to be to review the tubular cell cycle response upon acute kidney injury analysing novel targets for chronic kidney disease prevention. Overall, a comprehensive review of an important issue.

The article is very detailed and descriptive, and I enjoyed reading it and learning from it.

The topic is of great interest in the scientific community, the work is well organized and comprehensively described and the references are up to date. I also really appreciated the future perspective section.

I support the publication of this paper, but several aspects of this paper should be improved:

I suggest you improve the introduction section mentioning the role of vitamin D in tubular homeostasis as inflammation mediator, by the Megalin-Cubilin-Amnionless and FGF23-Klotho axis, you can find more information in the article “Gembillo G et al Protective Role of Vitamin D in Renal Tubulopathies. Metabolites. 2020 Mar19;10(3). pii: E115. doi: 10.3390/metabo10030115” and the role of vitamin d as immunomodulator in glomerulonephritis (see Gembillo G, Siligato R, Amatruda M, Conti G, Santoro D. Vitamin D and Glomerulonephritis. Medicina (Kaunas). 2021 Feb 22;57(2):186. doi: 10.3390/medicina57020186. PMID: 33671780; PMCID: PMC7926883).

We agree with the reviewer. We have expanded the introduction section and added one of the suggested reference (reference n. 21).

When you write about TGF-β, you should mention the relationship between AKI, elevated TGF-β and upregulation of the yes-associated protein (YAP)/transcriptional coactivator with PDZ-binding (TAZ) pathway, you can find more in the paper “Anorga S, Overstreet JM, Falke LL, Tang J, Goldschmeding RG, Higgins PJ, et al. Deregulation of Hippo-TAZ pathway during renal injury confers a fibrotic maladaptive phenotype. FASEB J. 2018;32(5):2644–57” and “Liang M, Yu M, Xia R, Song K, Wang J, Luo J, et al. Yap/Taz Deletion in Gli(+) Cell-Derived Myofibroblasts Attenuates Fibrosis. J Am Soc Nephrol. 2017;28(11):3278–90”.

We thank the reviewer for the suggestion. We have included the suggested information in the section 2.3.1. Lines 305-306, reference n. 95.

Another aspect that should be improved is the mitochondria dysfunction in AKI damage, you can find more in the paper “Yu SM, Bonventre JV. Acute kidney injury and maladaptive tubular repair leading to renal fibrosis. Curr Opin Nephrol Hypertens. 2020 May;29(3):310-318. doi: 10.1097/MNH.0000000000000605. PMID: 32205583; PMCID: PMC7363449” in the paragraph “Mitochondria Dysfunction”.

We thank the reviewer for the suggestion. We have included the suggested information in the introduction section. Lines 45-50.

A conclusion section should be added.

We added a conclusion section.

An abbreviations section should be added.

We added an abbreviation section.

A minor English check should be provided.

We fixed the typos and the typing errors throughout the manuscript.

The resolution of the images should be improved.

The resolution of the images has been improved.
